# Open access repository-scale propagated nearest neighbor suspect spectral library for untargeted metabolomics

Despite the increasing availability of tandem mass spectrometry (MS/MS) community spectral libraries for untargeted metabolomics over the past decade, the majority of acquired MS/MS spectra remain uninterpreted. To further aid in interpreting unannotated spectra, we created a nearest neighbor suspect spectral library, consisting of 87,916 annotated MS/MS spectra derived from hundreds of millions of MS/MS spectra originating from published untargeted metabolomics experiments. Entries in this library, or "suspects," were derived from unannotated spectra that could be linked in a molecular network to an annotated spectrum. Annotations were propagated to unknowns based on structural relationships to reference molecules using MS/MS-based spectrum alignment. We demonstrate the broad relevance of the nearest neighbor suspect spectral library through representative examples of propagation-based annotation of acylcarnitines, bacterial and plant natural products, and drug metabolism. Our results also highlight how the library can help to better understand an Alzheimer's brain phenotype. The nearest neighbor suspect spectral library is openly available for download or for data analysis through the GNPS platform to help investigators hypothesize candidate structures for unknown MS/MS spectra in untargeted metabolomics data.

When searching untargeted tandem mass spectrometry (MS/MS) metabolomics data using spectral libraries, on average only ~5% of the data can be annotated (~10% for human datasets)[1]. Unannotated spectra can arise due to incomplete coverage of the reference MS/MS spectral libraries of known compounds, including missing MS/MS spectra of different ion species, such as different ion forms, in-source fragments, and formation of multimers[2–4]. We hypothesized that many of the unidentified ions originate from different but related known molecules. Those molecules could be a result of host or microbial metabolism or promiscuous enzymes that accept various analogous substrates during biosynthesis[5]. To find related candidate ion species or to discover analogous MS/MS spectra from ions that originate from related molecules, strategies such as molecular networking[6] and other analog searching strategies[4,7–11] can be employed, for which molecular networking—a data visualization and interpretation strategy of MS/MS spectral alignment—in the Global Natural Products Social Molecular

Networking (GNPS) environment[12] is one of the most widely used tools[13].

In this work, we show that these strategies can also be used to generate new libraries of MS/MS reference spectra of potentially related MS/MS annotations from analog molecules that can subsequently be reused by the community. Previously, small reference spectral libraries of human milk oligosaccharides[14] and urine acylcarnitines[15] have been produced using an analog searching strategy (although the user licenses of these libraries restrict their redistribution). We hypothesized that the benefits of this approach could be further increased by considering analog matches across extremely large collections of MS/MS spectra to maximize the number of relevant spectrum links that can be found. Therefore, we have created a freely accessible and reusable MS/MS spectral library of MS/MS spectra related to identifiable molecules using molecular networking at the repository scale and created a nearest neighbor suspect spectral

✉ e-mail: wout.bittremieux@uantwerpen.be; pdorrestein@health.ucsd.edu

library to facilitate the annotation of mass spectrometry features that are present in public data.

## Results

### Nearest neighbor suspect spectral library creation

Using molecular networking, we have created a freely available and open-access mass spectral library of chemical analogs, referred to as the "nearest neighbor suspect spectral library." The library was created from compatible public datasets deposited to GNPS/MassIVE[12], MetaboLights[16], and Metabolomics Workbench[17]. In total, 521 million MS/MS spectra in 1335 public projects, with data from thousands of different organisms from diverse sources, including microbial culture collections, food, soil, dissolved organic matter, marine invertebrates, and humans, were used to compile the nearest neighbor suspect spectral library. Entries in this library, or "suspects," were derived from unannotated spectra that were linked in a molecular network (based on spectral similarity) to an annotated spectrum by MS/MS spectral library searching and where the precursor ion mass difference between the two spectra was non-zero.

A hierarchical processing strategy was employed to compile the nearest neighbor suspect spectral library from repository-scale public MS/MS data (Fig. 1). First, separate molecular networks were created for each dataset individually, while merging near-identical spectra and only keeping spectra that occur at least twice within the dataset to eliminate non-reproducible MS/MS spectra (Fig. 1, step 1). Spectrum annotations were obtained at the individual dataset level by matching against 221,224 reference spectra available in the GNPS community spectral libraries (June 2021) using parameters consistent with a false discovery rate $< 1\%$[12]. The cosine similarity was calculated using filtered spectra (the precursor $m/z$ peak was removed and only the top 6 most intense ions in every 50 $m/z$ window were included), and spectrum matches with a cosine score of 0.8 or higher and a minimum of 6 matching ions were accepted. Second, a global molecular network was created from all of the individual networks using the GNPS modified cosine similarity (Fig. 1, step 2). Finally, annotation propagations to the nearest neighbors were extracted from all molecular networks to create the library of nearest neighbor suspects (Fig. 1, step 3). To maximize the quality of the suspect annotations, suspects with infrequent mass offsets that occur fewer than ten times were excluded, as these are considered to be less-reproducible mass differences (Supplementary Fig. 1). Finally, a representative number of the annotation propagations were validated through expert manual inspection. The compilation of a global molecular network by co-networking thousands of datasets is an inherently more powerful strategy to discover

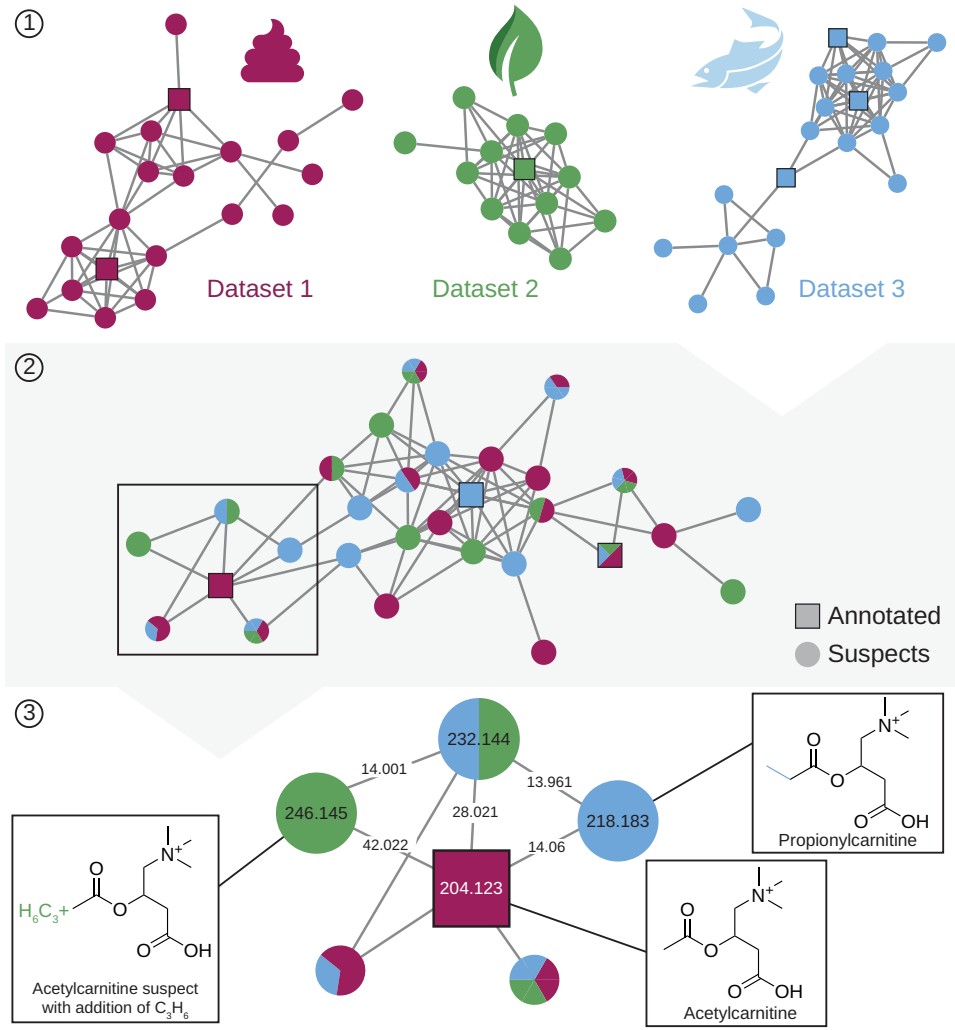

**Fig. 1 | Creation of the nearest neighbor suspect spectral library.** Overview of how the suspect library was created. Step 1: molecular networking of individual datasets. Step 2: co-networking of the 1335 datasets to create a global molecular network. Step 3: extract nearest neighbor suspects through annotation propagation to create the library.

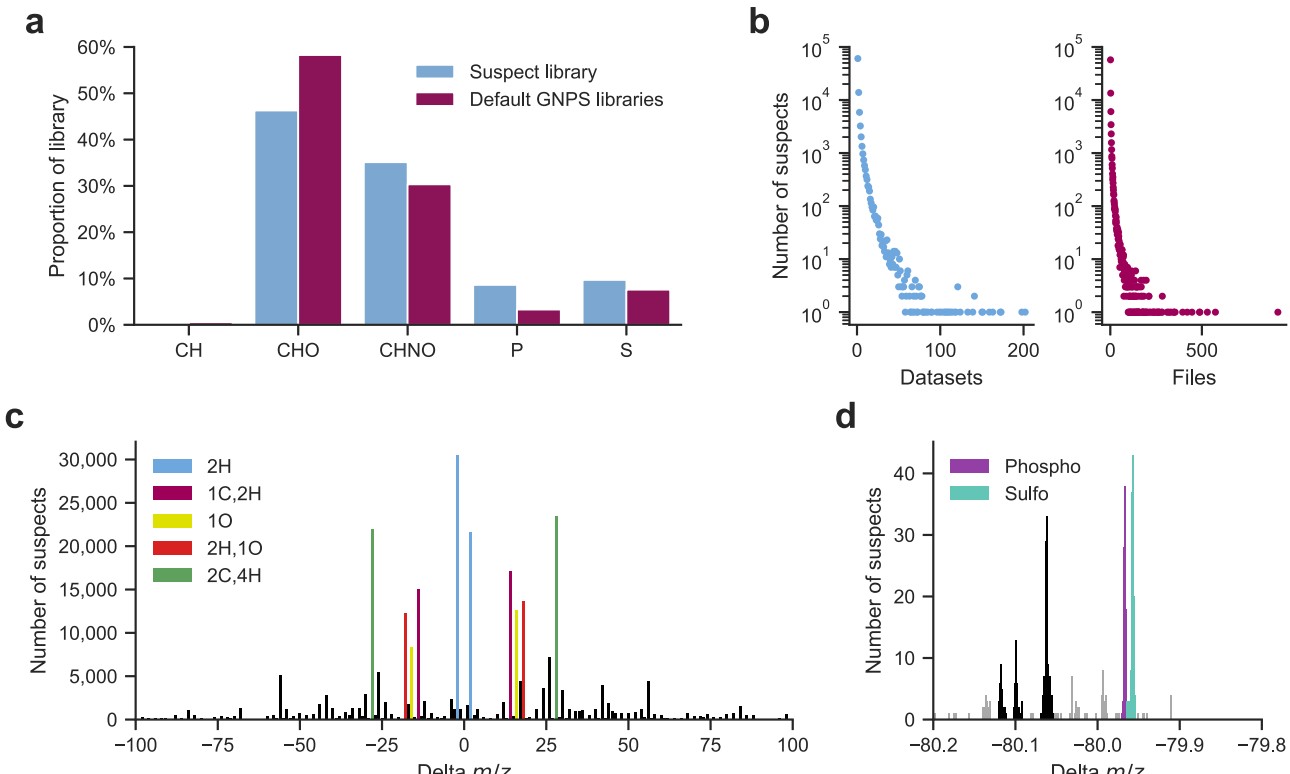

**Fig. 2 | Composition of the nearest neighbor suspect spectral library. a** The composition of suspects that exclusively exist of CH, CHO, CHNO, or contain P or S compared to the reference libraries. **b** Repeated occurrences of the suspects across datasets and files (i.e., individual LC-MS runs). **c** Frequently observed mass offsets (delta masses between pairs of spectra) associated with the suspect library. **d** Frequently observed mass offsets around a nominal mass of −80 Da. Source data are provided as a Source Data file.

relationships between MS/MS spectra, and thus between their corresponding molecules, than independent molecular networking within separate datasets, as moving to the repository scale makes it possible to discover patterns that cannot be detected from individual datasets in isolation[18]. For example, if molecules are transformed during metabolism, the unmodified form might only be present as an endogenous molecule in the originating organism, such as plant or animal-based food products, with a modified variant due to metabolism present in samples from humans that consumed these foods (Fig. 1).

In total, 87,916 unique MS/MS spectra and provenance to their matching analogs in the GNPS spectral libraries are included in the nearest neighbor suspect spectral library. Importantly, all of the nearest neighbor suspects are real spectra that occur in experimental data, whereas only a small portion (less than 10%) of reference MS/MS spectra in public and commercially available MS/MS spectral libraries have been observed in public data[1]. To homogenize and extend the information available for the suspects, molecular formulas were determined using SIRIUS[19] and BUDDY[20]. The elemental composition of the suspects reflects the characteristics of known reference libraries (Fig. 2a). For example, molecules that exclusively contain CH display poor ionization efficiency using electrospray ionization and are observed very rarely for both library types. Some suspects, such as common contaminants from sample vials, skin, or sodium formate clusters, as well as those related to endogenous molecules, such as fatty acids (e.g., vaccenic acid), bile acids (e.g., cholic acid), and lipids (e.g., phosphatidylcholines), are found in hundreds of public datasets and mass spectrometry files. In contrast, others, such as the natural products apratoxin, chelidonine, or marrubiin are observed less frequently (Fig. 2b, Supplementary Data 1).

There are 1350 frequent delta masses that occur in the nearest neighbor suspect spectral library (Fig. 2c, Supplementary Data 2).

When possible, the elemental composition of the delta masses and potential explanations, sourced from UNIMOD[21]—as many post-translational modifications or adducts that are observed in proteomics can also be found for small molecules—and a community-curated list of delta masses (Supplementary Data 3) are provided. The majority of delta mass explanations match the molecular formulas predicted by SIRIUS and BUDDY (Supplementary Fig. 2), indicating the complementarity of these approaches to interpret the structural modifications that the suspects have undergone. The most common mass offsets observed in the suspect library correspond to a gain or loss of 2.016 Da, which can be interpreted as the gain or loss of 2H (e.g., a double bond or ring structure), followed by a gain or loss of 28.031 Da, 14.016 Da, 18.011 Da, and 15.995 Da, corresponding to $C_2H_4$ (e.g., di(de)methylation or (de)ethylation), $CH_2$ (e.g., (de)methylation), $H_2O$ (e.g., water gain/loss), and O (e.g., (de)oxidation or (de)hydroxylation), respectively. However, 852 out of the 1350 mass offsets have not yet been explained (Supplementary Fig. 1). For example, although these mass offsets occur less frequently, there are at least five repeatedly observed offsets with a nominal delta mass of −80 Da (Fig. 2d), of which only phosphate loss (−79.966 Da) and sulfate loss (−79.957 Da) could currently be explained.

Spectral libraries are typically created by acquiring spectral data for pure standards, and reference MS/MS spectra have associated information on the precursor ions, compound names, and, when available, the molecular structures. In contrast, because the nearest neighbor suspect spectral library was compiled in a data-driven fashion, exact molecular structures are not known. Instead, the provenance of the suspect MS/MS spectra is described by their relationships to spectra that have an annotation, including the name and structure of the nearest neighbor MS/MS annotations and the observed pairwise delta masses. This is complemented by computed molecular formulas

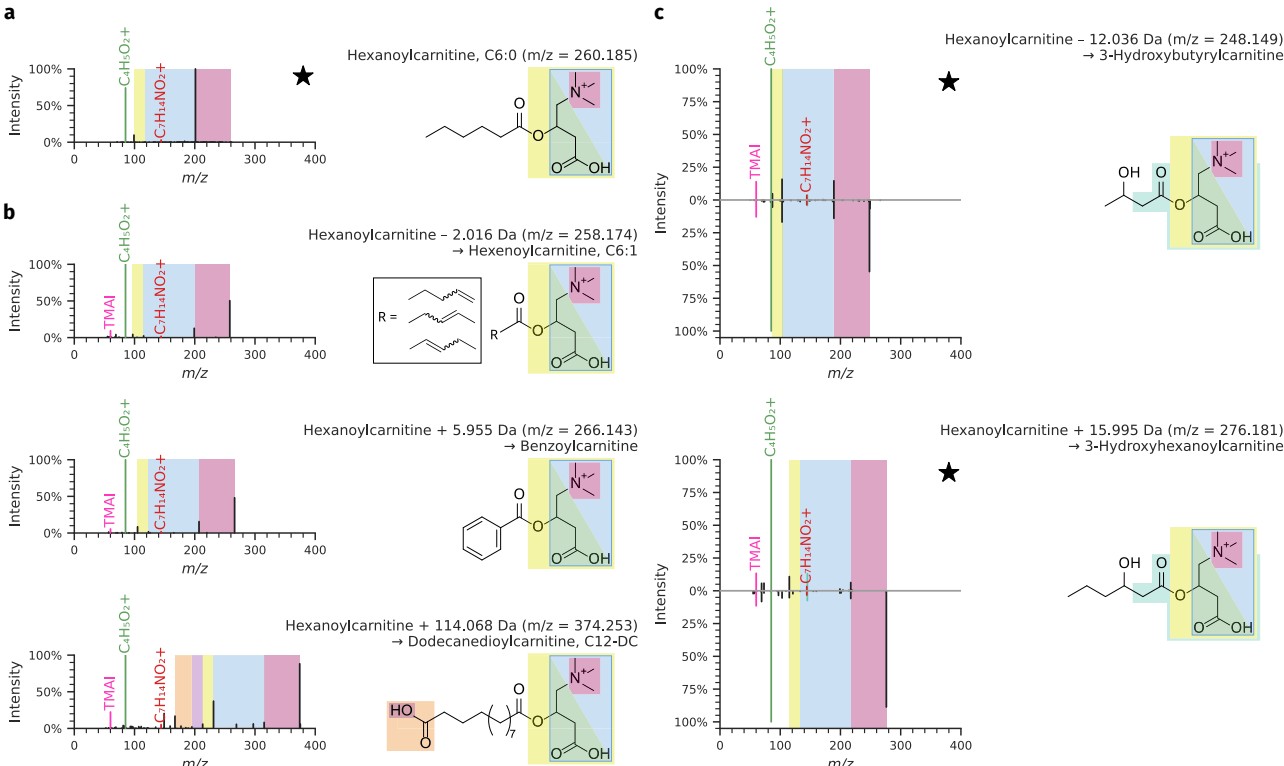

**Fig. 3 | Novel acylcarnitine reference spectra obtained using the nearest neighbor suspect spectral library.** Reference MS/MS spectra are indicated by ★. **a** Reference MS/MS spectrum for hexanoylcarnitine originally included in the GNPS community spectral libraries. **b** Nearest neighbor suspects related to hexanoylcarnitine. Annotations based on expert interpretation are: hexanoylcarnitine derived from a C6:1 fatty acid (unknown location of the double bond), benzoylcarnitine, and dodecanedioylcarnitine. **c** Nearest neighbor suspects related to hexanoylcarnitine for 3-hydroxybutyrylcarnitine and 3-hydroxyhexanoylcarnitine (bottom) confirmed against commercial standards (top). The suspect MS/MS spectra show a very high cosine similarity of 0.9988 and 0.9927 to the reference MS/MS spectra for 3-hydroxybutyrylcarnitine and 3-hydroxyhexanoylcarnitine, respectively.

and the elemental composition and potential explanation of the delta masses, as determined by matching against a curated list of delta masses. Suspects thus represent unknown molecules that are likely structurally related to reference molecules annotated using spectral library searching, with the location of the structural modification generally unspecified. Without any additional information, this is in agreement with a level 3 annotation (family level match) according to the Metabolomics Standards Initiative guidelines[22].

**Suspects provide structural hypotheses for observed molecules**
The nearest neighbor suspect spectral library covers various classes of molecules arising from both primary and specialized metabolism, including lipids, flavonoids, and peptides. A fundamental understanding of organic chemistry, mass spectral fragmentation, and awareness of the information that mass spectrometry can or cannot provide is key to achieve the deepest possible structural insights from the suspect library. To demonstrate how the nearest neighbor annotations can be used to propagate structural information, we highlight examples of acylcarnitines, apratoxin natural products, drug metabolism, flavonoids, and polymers in greater detail. Note that we do not discuss the stereochemistry of the suspect examples, as this information generally cannot be determined using mass spectrometry.

The first example involves several acylcarnitines, a group of molecules that plays a key role in mammalian—including human—energy cycling (Fig. 3)[23]. Hexanoylcarnitine, C6:0, is formed from the condensation of carnitine with hexanoic acid, a linear fatty acid with six carbons and zero double bonds (Fig. 3a). Based on expert interpretation of the MS/MS spectra for suspects related to hexanoylcarnitine, we were able to derive structural hypotheses for several related acylcarnitines (Fig. 3b). The first suspect example was initially annotated as

a hexanoylcarnitine but with a loss of 2.016 Da. This indicates that this suspect is likely a hexenoylcarnitine derived from a C6:1 fatty acid. Thus, the six carbon fatty acid tail now has one double bond, but the location of the double bond and its configuration (*E* vs *Z*) cannot be determined. The second acylcarnitine suspect was annotated as a hexanoylcarnitine with a gain of 5.955 Da, which corresponds to a gain of one C along with the loss of six hydrogens. The only structure that can match the acyl side chain is a planar benzoyl ester. The third acylcarnitine suspect example showed an addition of 114.068 Da, representing a carnitine with an acyl side chain that has two oxygens and twelve carbons, as derived from the mass difference and characteristic neutral losses for carnitine conjugates of dicarboxylic acids (179.121 Da and 207.130 Da)[24], which is consistent with dodecanedioylcarnitine. Additionally, we also found two suspects related to hexanoylcarnitine that include a characteristic 3-hydroxy fragment ion with a mass of 145.050 Da[25] (Fig. 3c). The first of these was initially annotated as hexanoylcarnitine but with a loss of 12.036 Da. Although close in value, based on accurate mass defects, the observed mass difference does not correspond to a loss of C (12.000 Da), but rather a combination that corresponds to the loss of $C_2H_4$ and gain of O. As the typical carnitine fragmentation pattern is conserved[15], we can determine that these changes occur in the fatty acid portion of the molecule, and thus, that this is likely a hydroxybutanoic acid carnitine derivative, leading to the final interpretation of 3-hydroxybutyrylcarnitine. The second 3-hydroxy suspect derived from hexanoylcarnitine showed an addition of 15.995 Da, representing a possible oxidation that can be localized based on the characteristic 3-hydroxy fragment[25], resulting in a spectrum annotation of 3-hydroxyhexanoylcarnitine. These two 3-hydroxy suspects also show excellent correspondence to commercial standards that were

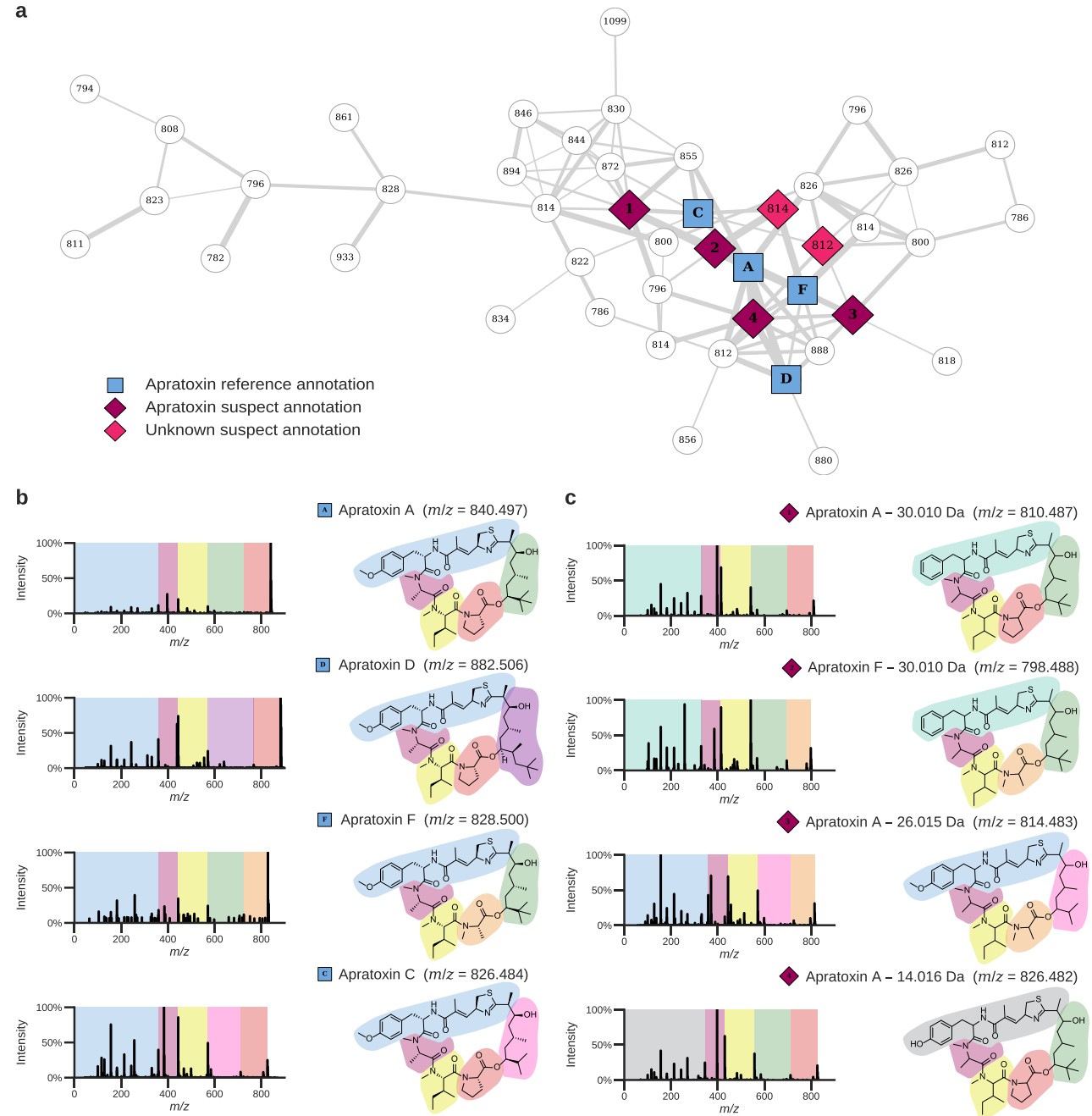

**Fig. 4 | Novel apratoxin reference spectra obtained using the nearest neighbor suspect spectral library. a** Apratoxin cluster in a molecular network created from *Moorena bouillonii* crude extracts. The reference spectral library hits are shown by the blue squares (**b**). The purple and pink diamond nodes represent matches to the nearest neighbor suspect spectral library, with the purple diamonds matching the MS/MS spectra shown for which structures could be proposed (**c**). The white nodes are additional MS/MS spectra within the apratoxin molecular family that remained unannotated, even when including the suspect library. **b** Reference MS/MS spectra and molecular structures of known apratoxins. **c** MS/MS spectra and structural hypotheses for four novel apratoxin suspects. All four apratoxin suspects were derived from the tropical marine benthic filamentous cyanobacterium *Moorena bouillonii*, which is known to produce apratoxins (MSV000086109 [https://doi.org/10.25345/C52475]).

subsequently purchased and measured, confirming our structural assignments (Fig. 3c).

The suspect library is also informative for the analysis of more complex molecules. The apratoxin family of natural products was isolated from filamentous cyanobacteria, and has been investigated in a number of biological systems due to its potent antineoplastic activities[26,27]. Using the suspect library to analyze a *Moorena bouillonii* cyanobacterial dataset achieved six additional spectrum annotations in the apratoxin molecular family (Fig. 4a). A structural annotation can be determined for four of these based on comparisons to the MS/MS spectra of apratoxin standards. The four MS/MS spectra in Fig. 4b show standards of purified apratoxin A, D, F, and C, while Fig. 4c shows four apratoxin suspects with proposed structures. Some key substitutions observed are proline for *N*-methylalanine, methoxytyrosine for tyrosine, and dimethyl versus trimethyl polyketide initiating units. These substitutions are likely generated due to biosynthetic promiscuity commonly associated with multimodular hybrid non-ribosomal peptide synthetases-polyketide synthases[28]. The apratoxin

suspects that were observed are apratoxin A and F with loss of 30.010 Da, apratoxin A with loss of 26.015 Da, apratoxin A with loss of 28.031 Da, and apratoxin A and F with loss of 14.016 Da; corresponding to $CH_2O$ (e.g., methoxy loss), $C_2H_2$ or $CH_2 + C$ loss, $C_2H_4$ loss (e.g., dimethylation), and $CH_2$ loss (e.g., methyl), respectively. The MS/MS spectra for four of the apratoxin suspects are shown in Fig. 4c. Based on the fragmentation, the $m/z$ difference corresponding to $CH_2$ loss in apratoxin A is due to unmethylated tyrosine, which could be explained by inactivity of an $O$-methyl transferase during biosynthesis. The fragmentation for both apratoxins with the 30.010 Da loss supports that the $m/z$ difference corresponding to methoxy loss is a result of phenylalanine incorporation by the associated adenylation domain rather than the methylated tyrosine observed in previously published apratoxin structures. Finally, although the loss of 26.015 Da is more complex, the other known apratoxins, together with their fragmentation, can be used to formulate a refined structural hypothesis. Compared to apratoxin A, the proline is likely substituted by an $N$-methylalanine, corresponding to a loss of C, and the trimethyl initiating unit is replaced by an isopropyl initiating unit. To obtain support for these modifications, isolation of this suspect (apratoxin A - 26.015 Da) was attempted from an extract of *Moorena bouillonii*; however, this resulted in a semi-pure fraction consisting of the suspect and small amounts of coeluting impurities. The semipure fraction was subjected to nuclear magnetic resonance (NMR) analysis (Supplementary Figs. 3–4). Compared to NMR analysis of apratoxin A and consistent with the mass spectrometry interpretation, the NMR correlations associated with proline are lost and the NMR signals corresponding to $N$-methylation of alanine are now observed. Substructure analysis based on the MS/MS data revealed that the polyketide synthase portion of the apratoxin suspect differs by one methyl group. This is consistent with the suspect containing an isopropyl group, as observed in apratoxin C, rather than the *tert*-butyl group observed in nearly all of the other apratoxins.

The suspect library also contains modified versions of known drugs that can arise due to in-source fragmentation, the formation of different ion species, incomplete synthesis or biosynthesis of the active ingredient that arises during manufacturing of the drug, or modifications introduced due to metabolism. An example is a suspect found in a human breast milk dataset matching the antibiotic azithromycin (Supplementary Fig. 5)[29]. The suspect is 14.015 Da lighter, consistent with a $CH_2$ loss. Based on inspection of the MS/MS data, it is possible to tentatively assign this loss of $CH_2$ to the methoxy group in the cladinose sugar, based on the presence of a hydroxy loss and absence of a methoxy loss.

Next, MS/MS data from medicinal plants listed in the Korean Pharmacopeia were analyzed using molecular networking. Several flavonoid diglycosides containing pentoses and hexoses were detected using MS/MS spectral library searching, with the default GNPS libraries providing ten spectrum annotations in this molecular family and the suspect library contributing annotations for 27 diglycoside analogs (Supplementary Fig. 6)[30]. Visual inspection of the MS/MS spectra indicated several modifications to formulate structural hypotheses for these suspects. For example, the apigenin-8-$C$-hexosylhexoside suspect with a delta mass of −30.019 Da corresponds to apigenin-8-$C$-pentosylhexoside. The presence of a pentose, instead of a hexose, is indeed consistent with the loss of $CH_2O$.

Finally, analysis of closely related polymeric substances resulted in a substantial increase in annotations (Supplementary Fig. 7). In an indoor chemistry environmental study[31], where a house was sampled before and after a month of human occupancy, there was a single spectrum match using the default GNPS libraries, to $p$-tert-octylphenol pentaglycol ether. Incorporating the suspect library added 55 matches that are related to polyethers, and that could be interpreted as part of a molecular family containing polymers. Thus, matching to the default GNPS spectral libraries alone gave the erroneous impression that there

were only a few octylphenol-polyethylene glycol molecules detectable within the house, while the suspect library revealed that there is a large and diverse group of them.

In conclusion, these examples highlight how annotations provided by MS/MS spectral libraries, including the nearest neighbor suspect spectral library, can assist in providing structural hypotheses at the molecular family level for observed molecules.

## Increases in MS/MS spectrum annotation provide new biomedical insights

To evaluate the spectrum annotation performance of the nearest neighbor suspect spectral library, we performed spectral library searching of public untargeted metabolomics data on GNPS/MassIVE (Fig. 5a). For the 1335 public datasets included during the creation of the suspect library, the default GNPS libraries resulted in an average MS/MS spectrum match rate of 5.5% (median 3.6%). Inclusion of the suspect library boosted the MS/MS spectrum match rate to 9.3% (median 6.4%), corresponding to 19 million additional spectrum matches. While these datasets were used to generate the suspect library, a similar increase in spectrum match rate was achieved for independent test data that were not part of the molecular networks from which the suspect library was compiled. For 72 datasets that were publicly deposited after the creation of the suspect library, the average spectrum match rate using the default GNPS libraries was 5.7% (median 4.7%), which increased to 8.9% (median 7.5%) when including the suspect library. Furthermore, we evaluated the performance of the suspect library for samples of different origins as recorded using the ReDU metadata system (Fig. 5b)[18]. For 45,845 raw files from 179 datasets with controlled vocabularies for sample information, such as animal (including human), bacterial, fungal, environmental, food, and plant samples, the suspect library consistently achieved an increased spectrum match rate, ranging from a $1.7 \pm 0.3$ fold increase in interpreted spectra for food data to $3.0 \pm 0.7$ fold increase for environmental samples (mean and standard deviation).

To further demonstrate the utility of the suspect library, we focused on untargeted metabolomics data from 514 human brains with and without Alzheimer's disease[32]. Using the default GNPS libraries there were 248,317 MS/MS spectral library matches, corresponding to 1305 unique molecule annotations. Including the suspect library increased the number of spectrum matches to 401,039, covering 5184 unique molecule annotations (Fig. 5c). One specific class of molecules that saw a particularly large increase in the number of annotations in this cohort was the acylcarnitines. There were 942 spectrum matches to 12 unique molecule annotations before the suspect library was included, but 1896 spectrum matches to 104 unique molecule annotations after inclusion of the suspect library.

We observed significant abundance differences for six acylcarnitines, as well as carnitine, when comparing brain metabolites between groups with and without Alzheimer's diagnosis (Fig. 5d). Three of those −carnitine, octanoylcarnitine, and lauroylcarnitine−could be annotated using the default GNPS libraries, while the remaining four metabolites could only be identified as acylcarnitines using the suspect library. The annotations of these seven metabolites were covered by three spectral library matches and eight suspect matches. When multiple spectra matched against the same metabolite, these annotations reinforced each other. For example, the carnitine annotation co-occurs with a match to the acetylcarnitine suspect with a loss of 42.010 Da. In this case, 42.010 Da corresponds to the mass of acetylation, which is lost in the suspect annotation, and therefore the suspect MS/MS spectrum represents carnitine itself. The four suspects that would otherwise remain unassigned as potential acylcarnitines are hexanoylcarnitine with loss of 12.036 Da (3-hydroxybutyrylcarnitine, Fig. 3c), hexanoylcarnitine with addition of 15.995 Da (3-hydroxyhexanoylcarnitine, Fig. 3c), decanoyl-L-carnitine with loss of 14.090 Da (-3C,-10H, + 2 O in the acyl chain), and decanoyl-L-carnitine

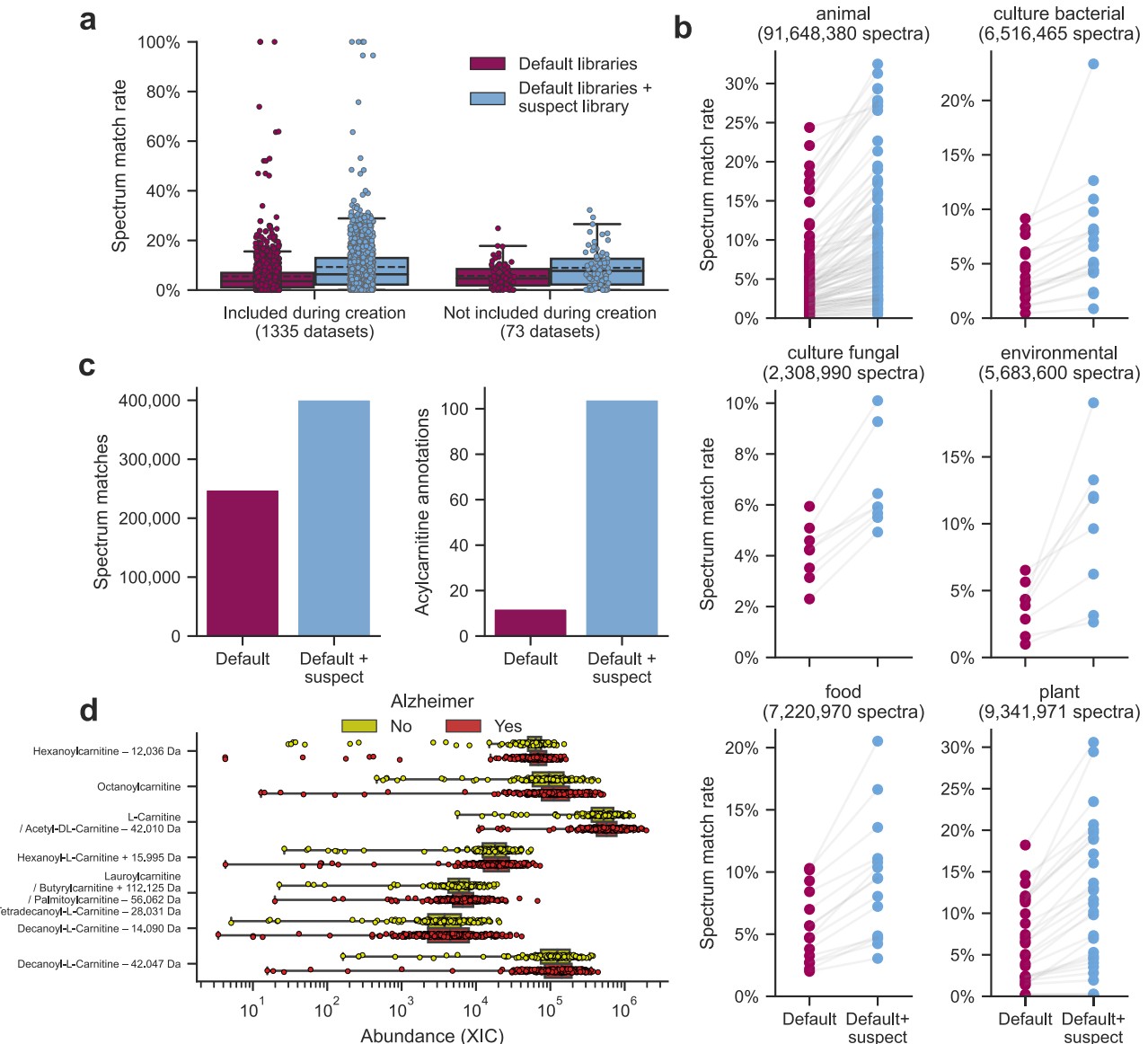

**Fig. 5 | Impact of the nearest neighbor suspect spectral library on spectrum matches to enable the formulation of structural hypotheses. a** The MS/MS spectrum match rate with and without the suspect library for 1407 public datasets on GNPS/MassIVE. The full center line indicates the median values, and the dashed center line indicates the mean values. The box limits indicate the first and third quartiles of the data, and the whiskers extend to 1.5 times the interquartile range. **b** The MS/MS spectrum match rate for different types of datasets with and without the suspect library. The data comes from 45,845 raw files in 179 datasets with known sample types recorded using the ReDU metadata system[18]. **c** MS/MS matches to an untargeted metabolomics human brain dataset from Alzheimer's disease patients ($n = 360$) and healthy subjects ($n = 154$) with and without the suspect library. **d** Differentially abundant carnitines for Alzheimer's disease patients ($n = 514$ biologically independent samples; Benjamini-Hochberg corrected p-value < 0.05). The suspect library was able to identify four additional mass spectrometry features as acylcarnitines, which would have remained unannotated matched only against the default GNPS libraries. Statistically significant carnitines were determined using

the Spearman correlations between all acylcarnitine extracted ion chromatograms (XICs) and the subjects' CERAD scores (a measure of Alzheimer progression, with 1 indicating "definite" Alzheimer's disease and 4 indicating "no" Alzheimer's disease) and the correlation coefficients and associated p-values were recorded. Multiple testing correction of the *p*-values was performed using the Benjamini-Hochberg procedure. For visualization purposes the four-scale CERAD score was binarized by considering a CERAD score of 1 or 2 to correspond to positive Alzheimer's disease patients, and a CERAD score of 3 or 4 to correspond to healthy individuals. The box limits indicate the first and third quartiles of the data, the center represents the median, and the whiskers extend to 1.5 times the interquartile range. *P*-values for the statistically significant carnitines are as follows. Hexanoylcarnitine − 12.036 Da → 0.00073; octanoylcarnitine → 0.03558; L-carnitine → 0.03966; hexanoyl-L-carnitine → 0.03966; lauroylcarnitine → 0.03966; decanoyl-L-carnitine − 14.090 Da → 0.03966; decanoyl-L-carnitine − 42.047 Da 0.03966. Source data are provided as a Source Data file.

with loss of 42.047 Da (-3C,-6H in the acyl chain). The first two suspects are related 3-hydroxy acylcarnitines that have 3-hydroxy-butyrate and 3-hydroxy-hexonate as the acyl side chain, whose predictions matched to data from commercial standards (Fig. 3c). The other two suspects are consistent with DC7:1 and C7:0 acylcarnitines[33]. These observations provide additional support that there are different fatty acids—now also including 3-hydroxy and odd-chain fatty acids—that are

transported as carnitine derivatives in Alzheimer's disease brains in comparison to healthy brains[34,35].

## Discussion

The annotation of untargeted metabolomics data is based on reference spectral libraries. However, because many known compounds and previously undiscovered analogs of compounds are unavailable as

reference standards, alternative approaches are required to interpret such MS/MS spectra. Here we have introduced a data-driven approach to compile an extensive nearest neighbor suspect spectral library. This library consists of 87,916 unique MS/MS spectra and can be freely downloaded as Mascot generic format and MSP files from the GNPS website. Additionally, through its direct integration in the spectral library searching and molecular networking functionality on the GNPS platform, the scientific community can incorporate the nearest neighbor suspect spectral library in their data analyses to formulate structural hypotheses.

The nearest neighbor suspect spectral library is closely related to analog searching. As suspects are derived from unannotated spectra that were linked in a molecular network to an annotated spectrum by MS/MS spectral library searching, conceptually analog searching can be used to directly annotate the MS/MS spectra as well. However, the nearest neighbor suspect spectral library has several ways it complements and/or brings advantages to analog searching. First, analog searching is computationally very expensive due to the massive search space that needs to be considered by opening up the precursor mass tolerance. As such, optimized algorithms[36,37] or even specialized hardware[38] are required to be able to do this efficiently. In contrast, the nearest neighbor suspect spectral library seamlessly works with standard spectral library searching procedures that are ubiquitously available. Second, analog searching suffers from an increased rate of false positive annotations, as random high-scoring matches are more likely to occur when considering a very large search space. In contrast, we explicitly safeguarded the quality of the nearest neighbor suspect spectral library by using stringent approaches towards spectrum matching and data filtering. Third, it can be challenging to interpret individual MS/MS spectrum annotations from analog searching, which generally involves a manual process. The nearest neighbor suspect spectral library addresses this by integrating relevant information from various sources and tools, including predicted molecular formulas and delta mass interpretations, to facilitate its annotations. In essence, this library contextualizes MS/MS spectrum annotations obtained from the nearest neighbor suspect spectral library within the full library and even the global molecular network from which it was compiled, making this an inherently more efficient strategy than analog searching.

Entries in the nearest neighbor suspect spectral library are not obtained by measuring pure reference standards. Therefore, it is important to consider that, initially, the exact molecular structure of the suspects is undetermined. Nevertheless, the suspect library includes essential information that can help to interpret MS/MS data that would otherwise remain entirely unexplored. Additionally, all of the spectra that are part of the suspect library have been detected experimentally and occur in biological data. In contrast, only a minority of the compounds contained in reference spectral libraries are actually observed in public data, indicating a mismatch between the laborious reference library creation efforts and the practical needs of metabolomics researchers. Consequently, incorporating the nearest neighbor suspect spectral library significantly increases the spectrum match rate across a wide variety of sample types. We have demonstrated how careful investigation of the suspects can provide highly detailed interpretations, and we anticipate that similar community contributions will be used to add and confirm further suspect annotations. Finally, when future studies uncover biologically relevant suspects, their molecular identities, including the location of modifications and stereochemical features, might be refined by measuring orthogonal properties, such as collision cross-section by ion mobility spectrometry or using genome mining, when possible. Ultimately, as is the case for all spectrum annotations, experimental validation of the complete molecular stereostructure requires either a reference standard or further isolation followed by structure elucidation by NMR, X-ray crystallography, or cryogenic electron microscopy experiments.

As the nearest neighbor suspect spectral library was compiled from hundreds of millions of MS/MS spectra deposited to public data repositories, its creation was only possible by building on the collective efforts of the scientific community over the past decade. Consequently, besides the tangible outcome of the library itself, an important contribution of this work is demonstrating the benefit of repository-scale data analysis and integration, by unlocking patterns and insights that individual studies, in isolation, could never reveal. This represents an important phase transition that now allows us to answer questions that were previously difficult to address. Furthermore, rather than a static resource, by harnessing the continuous data reanalysis efforts of the GNPS living data system[12] we will systematically incorporate newly deposited public data into the suspect library to continuously update its scope and applicability.

## Methods

### Integration of MetaboLights into GNPS/MassIVE

As a joint effort of the European Bioinformatics Institute (EMBL-EBI) and the GNPS/MassIVE teams, approximately 10,000 LC-MS/MS samples acquired in positive ion mode were imported from MetaboLights[16] into the GNPS/MassIVE repository by mirroring relevant files from MetaboLights on GNPS/MassIVE. These files represent over a hundred studies containing data from biologically diverse backgrounds, including but not limited to human, fungus, various bacterial and microbial species, and ecological samples. The data consist of both metabolomics and lipidomics samples.

### GNPS living data molecular networking

The nearest neighbor suspect spectral library was derived from molecular networking results as performed by GNPS's "living data" functionality, which periodically reanalyzes all publicly available untargeted metabolomics data on GNPS/MassIVE[12]. The living data analysis (update performed on November 17, 2020) includes results for 1335 datasets, corresponding to 520,823,130 million MS/MS spectra. Spectrum clustering grouped 168,193,526 MS/MS spectra in 8,543,020 clusters while discarding 352,629,604 singleton spectra, of which 454,091 cluster representatives could be annotated using spectral library searching against the default GNPS spectral libraries (5.3% annotation rate), and which formed a molecular network consisting of 13,179,147 spectrum pair edges.

This analysis consisted of two phases of spectrum clustering using MS-Cluster[39] and molecular networking. First, spectra were networked within each individual dataset. Per-dataset molecular networking outputs are available on the MassIVE repository with dataset identifier MSV000084314 [https://doi.org/10.25345/C5WQ0T]. Next, a second round of molecular networking was performed on the combined consensus spectra for all datasets generated from the first molecular network.

Spectra were preprocessed by removing all MS/MS fragment ions within +/- 17 Da of the precursor $m/z$. Only the top 6 most abundant ions in every 50 $m/z$ window were retained. The first round of molecular networking used a precursor mass tolerance of 2.0 $m/z$, a fragment mass tolerance of 0.5 $m/z$, and three rounds of MS-Cluster clustering with mixture probability threshold 0.05. Thresholds for the second round of molecular networking were modified due to computational and memory constraints, and consisted of a precursor mass tolerance of 0.1 $m/z$ and a fragment mass tolerance of 0.1 $m/z$. MS-Cluster used the standard cosine similarity to group near-identical MS/MS spectra, whereas the molecular networking analyses employed a GNPS modified cosine similarity that takes directly matching ions into account as well as ions that are shifted according to the precursor mass difference[10]. The molecular networking used a minimum cosine similarity of 0.8, minimum six matched peaks, only considered clusters that consist of at least two MS/MS spectra, and retained the ten strongest edges for each node in the molecular network.

Spectral annotations were obtained through spectral library searching against the default GNPS spectral libraries (GNPS Collections Bile Acid Library 2019[12], CASMI[40], Dereplicator Identified MS/MS Spectra[41], GNPS Collections Miscellaneous[12], Pesticides, EMBL Metabolomics Core Facility[42], Faulkner Legacy Library provided by Sirenas MD, GNPS Library[12], NIH Clinical Collection 1, NIH Clinical Collection 2, NIH Natural Products Library Round 1[43], NIH Natural Products Library Round 2[43], Pharmacologically Active Compounds in the NIH Small Molecule Repository, GNPS Matches to NIST14[12], PhytoChemical Library, FDA Library Pt 1, FDA Library Pt 2, HMDB[44], LDB Lichen Database[45], Massbank Spectral Library[46], Massbank EU Spectral Library, MIADB Spectral Library[47], Medicines for Malaria Venture Pathogen Box, Massbank NA Spectral Library, Pacific Northwest National Lab Lipids[48], ReSpect Spectral Library[49], Sumner/Bruker), which contained 221,224 reference MS/MS spectra (June 2021). Settings for the living data spectral library searching step included a precursor ion tolerance of 2.0 $m/z$, a fragment ion tolerance of 0.5 $m/z$, a minimum cosine similarity of 0.7, and minimum six matched peaks.

### Nearest neighbor suspect spectral library creation

High-quality MS/MS spectra were extracted from the GNPS living data molecular network to compile the nearest neighbor suspect spectral library. Suspects were derived from spectrum pairs for which only one of the spectra was identified during spectral library searching and both spectra have a non-zero precursor mass difference. In this case, the unidentified spectrum was included in the nearest neighbor suspect spectral library, as it corresponds to a previously unknown molecule that is structurally related to the reference molecule identified using spectral library searching. Strict filtering thresholds were used to avoid inclusion of incorrect entries: spectrum–spectrum matches required a maximum precursor mass tolerance of 20 ppm, a minimum cosine similarity of 0.8, and minimum six matched ions.

To homogenize and extend the information available for the suspects, their molecular formulas were determined using SIRIUS[19] and BUDDY[20]. For the SIRIUS (version 4.5.2) analysis, only MS/MS data were used as input and the precursor mass tolerance was set to 10 ppm for Orbitrap spectra and 25 ppm for Q-TOF spectra. For the BUDDY (version 1.3) analysis, a precursor mass tolerance of 5 ppm for FT-ICR spectra, 10 ppm for Orbitrap spectra, and 25 ppm for Q-TOF spectra was used. The fragment mass tolerance was set to twice the precursor mass tolerance and no database restriction was applied. All other settings were kept at their default values. If BUDDY could not annotate MS/MS spectra with molecular formulas while only considering CHNOPS elements, the spectra were subsequently reannotated while considering additional elements: CHNOPSFClBrI. Molecular formulas predicted by SIRIUS and BUDDY agreed for 32,302 suspects, while SIRIUS predicted a different molecular formula for 32,508 suspects and BUDDY for 47,508 suspects. In case SIRIUS and BUDDY predicted different molecular formulas, both were included in the suspect name (see below).

Additionally, the observed precursor mass differences were calibrated and matched to putative modification explanations contained in the UNIMOD database[21] and a manually compiled list of modifications and their mass differences (Supplementary Data 3). Suspects whose delta mass occurred fewer than ten times were discarded, as true modifications are expected to occur repeatedly for different molecules, while suspects with infrequent mass differences more likely correspond to spurious matches.

To ensure that the provenance of the suspects to the matched reference molecules compared to which they are annotated based on spectral similarity is properly understood, their names are of the form: "Suspect related to [*compound name*] (predicted molecular formula: [*molecular formula SIRIUS and/or BUDDY*]) with delta $m/z$ [*positive (addition) or negative (loss) delta m/z*] (putative explanation: [*modification*])." In case multiple propagations to different reference spectra are available, information for all matches is included.

### Spectrum annotation using the nearest neighbor suspect spectral library

The spectrum annotation performance of the nearest neighbor suspect spectral library was assessed by large-scale spectral library searching using the default GNPS spectral libraries excluding and including the nearest neighbor suspect spectral library on 1407 public datasets available on GNPS/MassIVE, consisting of a combined 592 million MS/MS spectra. Of these datasets, 1335 datasets were also included in the GNPS living data analysis from which the nearest neighbor suspect spectral library was compiled (521 million MS/MS spectra; see above) and 72 datasets were deposited at a later date and can be considered a completely independent test set (72 million MS/MS spectra). All searches used a precursor mass tolerance of 2.0 $m/z$, a fragment mass tolerance of 0.5 $m/z$, a minimum cosine similarity of 0.8, and minimum 6 matched peaks. Other options were kept at their default values.

### Evaluation of acylcarnitine suspects

**Mass spectrometry analysis.** Structural hypotheses for several suspect acylcarnitines were confirmed using reference standards based on spectral matches, accurate masses, and retention times: [(3 R)-3-Hydroxybutyryl]-L-carnitine (Catalog No. 918639-76-6, Sigma-Aldrich Inc.), [(3 R)-3-Hydroxyhexanoyl]-L-carnitine (Catalog No. 1469900-93-3, Sigma-Aldrich Inc.) and Hexanoyl-L-carnitine (Catalog No. 22671-29-0, Sigma-Aldrich Inc.). Standards were prepared at 1 µM concentration. Untargeted LC-MS/MS acquisition was performed on a Vanquish Ultrahigh Performance Liquid Chromatography (UHPLC) system coupled to a Q-Exactive Hybrid Quadrupole-Orbitrap (Thermo Fisher Scientific, Bremen, Germany). Chromatographic separation was performed on a Kinetex 1.7 µm 100 Å pore size C18 reversed phase UHPLC column 50 × 2.1 mm (Phenomenex, Torrance, CA) with a constant flow rate of 0.5 mL/min. The following solvents were used during the LC-MS/MS acquisition: water with 0.1% formic acid (v/v), Optim LC/MS grade, Thermo Scientific (solvent A) and acetonitrile with 0.1% formic acid (v/v), Optima LC/MS grade, Thermo Scientific (solvent B). After injection of 1 µL of sample into the LC system and eluted with isocratic gradient of 5% B from 0 to 1 min and linear gradient from 5 to 100% B (1–7 min), 100% B (7–7.5 min), 100 to 5% B (7.5–8 min), 5% B (8–10 min). Data dependent acquisition mode was used for acquisition of MS/MS data with default charge state of 1. An inclusion list containing the following ions was used: $m/z$ 260.18563 (molecular formula: C13H25NO4, start: 2.00 min, end: 3.00 min), $m/z$ 248.14925 (molecular formula: C11H21NO5, start: 0.00 min, end: 1.00 min), $m/z$ 276.18055 (molecular formula: C13H25NO5, start: 0.50 min, end: 1.50 min). Full MS was acquired using 1 microscan at a resolution of 35,000 at 200 $m/z$, automatic gain control (AGC) target 5e5, maximum injection time of 100 ms, scan range 100–1500 $m/z$ and data acquired in profile mode. DDA of MS/MS was acquired using 1 microscan at a resolution of 35,000 at 200 $m/z$, AGC target 5e5, top 5 ions selected for MS/MS with isolation window of 2.0 $m/z$ with scan range 200–2000 $m/z$, fixed first mass of 50 $m/z$ and stepped normalized collision energy of 20, 30, and 40 eV, minimum AGC target 5e3, intensity threshold 5e4, apex trigger 2 to 15 s, all multiple charges included, isotopes were excluded, and a dynamic exclusion window of 10 s.

### Evaluation of apratoxin suspects

**Mass spectrometry analysis.** Apratoxin suspects were investigated in the context of *Moorena bouillonii*, a tropical marine benthic filamentous cyanobacterium. The mass spectrometry data were derived from both field-collected and laboratory-cultured biomass of *Moorena bouillonii* (MassIVE dataset identifier MSV000086109 [https://doi.org/10.25345/C52475]). A number of collections are represented in this

dataset, including those originating from sites around Guam, Saipan (Commonwealth of the Northern Mariana Islands), Palmyra Atoll, Papua New Guinea, American Samoa, Kavaratti (Lakshadweep, India), the Paracel Islands (Xisha, China), the Solomon Islands, and the Red Sea (Egypt). The biomass from each of the samples was extracted using 2:1 dichloromethane and methanol. The crude extracts were concentrated and resuspended in acetonitrile, followed by a desalting protocol using C18 SPE with acetonitrile. Samples were then resuspended in methanol containing 2 μM sulfamethazine as an internal standard. Untargeted metabolomics was performed using an UltiMate 3000 liquid chromatography system (Thermo Scientific) coupled to a Maxis Q-TOF (Bruker Daltonics) mass spectrometer with a Kinetex C18 column (Phenomenex). Data were collected in positive ion mode using data-dependent acquisition. All solvents used were LC-MS grade.

**Molecular networking.** Molecular networking and spectral library searching were performed using the GNPS platform as described above. Settings included a precursor mass tolerance of 2.0 $m/z$, fragment mass tolerance of 0.5 $m/z$, minimum cosine similarity of 0.7, and minimum 6 matched peaks. Data visualization was performed using the Metabolomics USI interface[50] and spectrum–spectrum matches were evaluated manually to develop hypotheses regarding the structure of apratoxin analogs that were annotated using the nearest neighbor suspect spectral library.

**Cyanobacterial culture.** *Moorena bouillonii* PNG5-198 was initially collected by scuba in 3–10 m of water off the coast of Pigeon Island, Papua New Guinea (S4 16.063′ E152 20.266′) in May 2005. Live cultures have been maintained in SWBG-11 media under laboratory conditions at 27 °C and a 16/8 h light/dark schedule. Biomass for *Moorena bouillonii* was obtained through ongoing laboratory culture.

**Extraction and isolation of apratoxins.** The cultured biomass was extracted using 2:1 $CH_2Cl_2$/MeOH affording 241.4 mg of organic extract. The extract was then subjected to vacuum liquid chromatography (VLC) on silica gel (type H, 10–40 μm) using normal phase solvents in a stepwise gradient of hexanes/EtOAc and EtOAc/MeOH, resulting in nine fractions (A-I). The fraction eluting with 25% MeOH/75% EtOAc (fraction H) had a mass of 21.6 mg. This fraction was found to have the characteristic MS/MS signatures of the apratoxins and was selected for further purification using reversed-phase HPLC. A Phenomenex Kinetex C18 5μm 100 Å 100 × 4.6 mm column with a 3 mL/min was used to obtain 1.2 mg of semipure suspect (apratoxin A - 26.015 Da) and 2.1 mg of semipure apratoxin A.

**NMR spectroscopy.** 1H NMR and 2D NMR spectra were obtained on a Bruker Advance III DRX-600 NMR with a 1.7 mm dual tune TCI cryoprobe (600 MHz and 150 MHz for $^1$H and $^{13}$C, respectively). NMR spectra were referenced to residual solvent $CDCl_3$ signals as an internal standard. NMR spectra were processed using MestReNova (Mnova 14.2.3, Mestrelab Research).

### Evaluation of azithromycin suspects
**Mass spectrometry analysis.** The presence of azithromycin suspects was investigated using human breast milk data (MassIVE dataset identifier MSV000081432 [https://doi.org/10.25345/C58K7570Q])[29]. Human milk samples were extracted using 80:20 methanol and water. Untargeted metabolomics was performed using an UltiMate 3000 liquid chromatography system (Thermo Scientific) coupled to a Maxis Q-TOF (Bruker Daltonics) mass spectrometer with a Kinetex C18 column (Phenomenex). Samples were run using a linear gradient of mobile phase A (water 0.1% formic acid (v/v)) and phase B (acetonitrile 0.1% formic acid (v/v)). A representative linear gradient consisted of 0–0.5 min isocratic at 5% B, 0.5–8.5 min 100% B, 8.5–11 min isocratic at 100% B, 11–11.5 min 5% B, and 11.5–12 min 5% B. Data were collected in positive ion mode using data-dependent acquisition. All solvents used were LC-MS grade.

**Molecular networking.** Molecular networking and spectral library searching were performed using the GNPS platform as described above. Settings included a precursor mass tolerance of 0.02 $m/z$, fragment mass tolerance of 0.02 $m/z$, minimum cosine similarity of 0.6, and minimum 5 matched peaks. Data visualization was performed using the Metabolomics USI interface[50] and spectrum–spectrum matches were evaluated manually to interpret the azithromycin suspect.

### Evaluation of flavonoid suspects
**Mass spectrometry analysis.** Untargeted metabolomics data for medicinal plants listed in the Korean Pharmacopeia were used to investigate flavonoid suspects (MassIVE dataset identifier MSV000086161 [https://doi.org/10.25345/C5SB50]). Samples were extracted using methanol. Untargeted metabolomics was performed using an Acquity liquid chromatography system coupled to a Xevo G2 Q-TOF (Waters) mass spectrometer with a BEH C18 column at 40 °C (Waters Corp.; 50 mm; 2.1 mm; 1.7 μm particle size). Water (solvent A) and acetonitrile (solvent B) were used as mobile phase, both with 0.1% formic acid, and a method of 20 min (linear gradient), flow 0.3 mL/min was performed using the following settings: 0–14 min. from 5 to 95% B; 14–17 min, 95% B; 17–17.1 min from 95% to 5% B; 17.1–20 min, 5% B for equilibration of the column for the next sample. Data were collected in positive and negative ion modes using data-dependent acquisition. All solvents used were LC-MS grade.

**Molecular networking.** Molecular networking and spectral library searching were performed in the GNPS platform as described above. Settings included a precursor ion mass tolerance of 2.0 $m/z$, fragment ion mass tolerance of 0.5 $m/z$, minimum cosine similarity of 0.7, and minimum 6 matched peaks. Data visualization was performed using the Metabolomics USI interface[50] and spectrum–spectrum matches were evaluated manually to interpret the flavonoid suspects.

### Home environment personal care products
**Mass spectrometry analysis.** The presence of polymeric suspects was investigated in the context of the HOMEChem project, a study of the indoor chemical environment (MassIVE dataset identifier MSV000083320 [https://doi.org/10.25345/C5CP63])[31]. For full details on the experimental set-up, see Aksenov et al. (2021)[31]. Briefly, scripted activities, including cleaning and cooking, were performed in a controlled home environment. Sample collection consisted of swabbing different locations in the test house. Untargeted metabolomics was performed using a Vanquish liquid chromatography system (Thermo Scientific) coupled to a QExactive Orbitrap (Thermo Scientific) mass spectrometer with a Kinetex C18 column (Phenomenex). The mobile phase used was water (phase A) and acetonitrile (phase B), both containing 0.1% formic acid (Fisher Scientific, Optima LC/MS), employing the following gradient: 0–1 min 5% B, 1–8 min 100% B, 8–10.9 min 100% B, 10.9–11 min 5% A, 11–12 min 5% B. Data were collected in positive ion mode using data-dependent acquisition. All solvents used were LC-MS grade.

**Molecular networking.** Molecular networking and spectral library searching were performed using the GNPS platform as described above. Settings included a precursor mass tolerance of 0.02 $m/z$, fragment mass tolerance of 0.02 $m/z$, minimum cosine similarity of 0.7, and minimum 6 matched peaks.

### Alzheimer's disease acylcarnitine analysis
**Mass spectrometry analysis.** The presence of acylcarnitine suspects was investigated in the context of the Religious Orders Study/Memory and Aging Project (ROSMAP) to study Alzheimer's disease

(MassIVE dataset identifier MSV000086415 [https://doi.org/10.25345/C5977C][32]. Untargeted metabolomics was performed on human brain samples from 514 individuals with and without Alzheimer's disease (360 Alzheimer's disease patients, 154 healthy subjects). Human brain tissue samples were placed into tubes with 800 μl of a 1:1 mixture of $H_2O$ (Optima LC-MS grade W64) and MeOH (100%) containing 1 μM of sulfamethazine. The samples were homogenized using a Qiagen TissueLyser II at 25 Hz for 5 min, then centrifuged at 14,000 g for 5 min before being incubated for a period of 30 min at −20 °C. A 200 μl aliquot of supernatant from each sample was transferred into a 96-well plate and vacuum concentrated to dryness via centrifugal lyophilization (Labconco Centrivap). Once dried, the samples were stored at −80 °C until LC-MS was performed. Untargeted metabolomics was performed using a Vanquish liquid chromatography system (Thermo Scientific) coupled to a QExactive (Thermo Scientific) mass spectrometer with a C18 column (Phenomenex Kinetex 1.7 μm C18 100 Å LC Column 50 × 2.1 mm). The mobile phase used was LC-MS grade water (phase A) and LC-MS grade acetonitrile (phase B), both containing 0.1% formic acid (Fisher Scientific, Optima LC-MS), with a flow rate set to 0.5 mL/min. Samples were injected at 95%A:5%B, which was held for 1 min, before ramping up to 100%B over 7 min, which was held for 0.5 min before returning to starting conditions. Data were collected in positive ion mode using data-dependent acquisition to acquire MS full scan spectra, followed by MS/MS spectra of the top 5 most abundant ions. Precursor ions were fragmented once before being added to an exclusion list for 30 s.

**Data analysis.** Spectral library searching was performed using the GNPS platform as described above using the default GNPS spectral libraries only and including the nearest neighbor suspect spectral library. Settings included a precursor mass tolerance of 2.0 $m/z$, fragment mass tolerance of 0.5 $m/z$, minimum cosine similarity of 0.8, and minimum 6 matched peaks. Raw MS data visualization was performed using the GNPS Dashboard[51]. Spectrum annotations corresponding to carnitines were extracted by filtering on "carnitine" in the compound name. Different spectrum annotations with near-identical precursor $m/z$ (precursor $m/z$ tolerance 100 ppm) and retention time (retention time tolerance 20 s) were merged. Feature abundances were obtained by computing extracted ion chromatograms (XICs) with $m/z$ tolerance 100 ppm and retention time tolerance 20 s for all uniquely annotated acylcarnitines across all 514 raw files. Next, the Spearman correlations between all acylcarnitine XICs and the subjects' CERAD scores (a measure of Alzheimer's disease progression, with 1 indicating "definite" Alzheimer's disease and 4 indicating "no" Alzheimer's disease) were calculated and the correlation coefficients and associated $p$-values were recorded. Multiple testing correction of the $p$-values was performed using the Benjamini-Hochberg procedure, and acylcarnitines with a corrected $p$-value below 0.05 were considered to be significantly associated with Alzheimer's disease. For visualization purposes the four-scale CERAD score was binarized by considering a CERAD score of 1 or 2 to correspond to positive Alzheimer's disease patients, and a CERAD score of 3 or 4 to correspond to healthy individuals.

**Reporting summary**
Further information on research design is available in the Nature Portfolio Reporting Summary linked to this article.

## Data availability
Source data are provided with this paper. All of the data involved in this work are publicly available through GNPS/MassIVE: GNPS living data molecular networking. • GNPS living data (version November 17, 2020) [https://gnps.ucsd.edu/ProteoSAFe/result.jsp?task=25cc4f9135c6428aabe1f41a9e54c369&view=advanced_view]. • Living data global molecular network [https://gnps.ucsd.edu/ProteoSAFe/status.jsp?task=4f 69e11bfb544010b2c4225a255f17ba]. Spectrum annotation using the nearest neighbor suspect spectral library. • Spectral library searching using the default GNPS libraries only. ○ part 1 [https://gnps.ucsd.edu/ProteoSAFe/status.jsp?task=308b3393a2b2401e8c9b562152531b4c]. ○ part 2 [https://gnps.ucsd.edu/ProteoSAFe/status.jsp?task=18cf4e52 1f9b4124af54d7e3d837a888]. ○ part 3 [https://gnps.ucsd.edu/ProteoSAFe/status.jsp?task=c0249eb6a52e4ea993b03de90a509b35]. ○ part 4 [https://gnps.ucsd.edu/ProteoSAFe/status.jsp?task=debd3bbb51f64 90394e905e13779f295]. part 5 [https://gnps.ucsd.edu/ProteoSAFe/status.jsp?task=8cdb4d7d1a784f5bb4f99e4c31564cd1]. ○ part 6 [https://gnps.ucsd.edu/ProteoSAFe/status.jsp?task=a9e7e4b1b810441 6a39142fd6072e02a]. ○ part 7 [https://gnps.ucsd.edu/ProteoSAFe/status.jsp?task=334ed0d944844e90b71d6151d4e74263]. ○ part 8 [https://gnps.ucsd.edu/ProteoSAFe/status.jsp?task=b55aef34c0bd4d7 8a1f3952f7c49a52c]. • Spectral library searching using the default GNPS spectral libraries and the nearest neighbor suspect spectral library. ○ part 1 [https://gnps.ucsd.edu/ProteoSAFe/status.jsp?task= 064be855f46e407f9f5fcbe652c8b9d5]. ○ part 2 [https://gnps.ucsd.edu/ProteoSAFe/status.jsp?task=d243afb8f233490886bb8ab5eedcf8 b8]. ○ part 3 [https://gnps.ucsd.edu/ProteoSAFe/status.jsp?task= febab54db7a14af6b451ab5e5789785f]. ○ part 4 [https://gnps.ucsd.edu/ProteoSAFe/status.jsp?task=eba0dfe63a464b0a924fd5e373917b 37]. ○ part 5 [https://gnps.ucsd.edu/ProteoSAFe/status.jsp?task=95b 541cb3be54d08a0b14367554630ca]. ○ part 6 [https://gnps.ucsd.edu/ProteoSAFe/status.jsp?task=1df48f2dc7c443fc9364dfc8b28f6b47]. ○ part 7 [https://gnps.ucsd.edu/ProteoSAFe/status.jsp?task=b7f8c3d 47a464b53ab94f1780f56c893]. ○ part 8 [https://gnps.ucsd.edu/ProteoSAFe/status.jsp?task=50e3d8ae4e004f989862fcc9d1353534]. Evaluation of suspect use cases. • Molecular networking of apratoxin suspects [https://gnps.ucsd.edu/ProteoSAFe/status.jsp?task=5c4169 3f607d4b4cabbcfbbf5b9bcf86]. • Molecular networking of azithromycin suspects [https://gnps.ucsd.edu/ProteoSAFe/status.jsp?task=e91e2e44e3234f08bb3d7f3f16d5f782]. • Molecular networking of flavonoid suspects [https://gnps.ucsd.edu/ProteoSAFe/status.jsp?task=38a1bd60bd094c8a97cf49d822e7f853]. • Molecular networking of home environment personal care products [https://gnps.ucsd.edu/ProteoSAFe/status.jsp?task=890e39f28140470ab0598c77cc5c048e]. • Spectral library searching of Alzheimer's disease data. ○ Using the default GNPS spectral libraries only [https://gnps.ucsd.edu/ProteoSAFe/status.jsp?task=b55aef34c0bd4d78a1f3952f7c49a52c]. ○ Using the default GNPS spectral libraries and the nearest neighbor suspect spectral library [https://gnps.ucsd.edu/ProteoSAFe/status.jsp?task=50e3d8ae4e004f989862fcc9d1353534]. Additionally, all relevant data files have been deposited to a permanent Zenodo archive at https://doi.org/10.5281/zenodo.8282733. Potential explanations for the observed delta masses were partially sourced from the UNIMOD database of protein modifications (https://www.unimod.org/)[21], complemented with a community-curated list of delta masses (Supplementary Data 3). Metabolomics and clinical data for the ROSMAP clinic cohorts are also available via the AD Knowledge Portal (https://adknowledgeportal.org) and through request to Dr. David Bennett at Rush University who provided brain samples used for analysis. The AD Knowledge Portal is a platform for accessing data, analyses, and tools generated by the Accelerating Medicines Partnership (AMP-AD) Target Discovery Program and other National Institute on Aging (NIA)-supported programs to enable open-science practices and accelerate translational learning. The data, analyses, and tools are shared early in the research cycle without a publication embargo on secondary use. Data is available for general research use according to the following requirements for data access and data attribution (https://adknowledgeportal.synapse.org/Data%20Access). For access to content described in this manuscript see: https://doi.org/10.7303/syn30255033.1. The nearest neighbor suspect spectral library is freely available under the CC0 license at https://gnps.ucsd.edu/ProteoSAFe/gnpslibrary.jsp?library=GNPS-SUSPECTLIST and archived

on Zenodo at https://doi.org/10.5281/zenodo.8282733. Additionally, it can be used for any data analysis task on GNPS by selecting it from the CCMS_SpectralLibraries > GNPS_Propogated_Libraries > GNPS-SUS-PECTLIST > GNPS-SUSPECTLIST.mgf path in the GNPS file selector dialog. Step-by-step instructions are also provided on GitHub at https://github.com/bittremieux/gnps_suspect_library and on the GNPS Documentation website. Individual spectra are accessible by their Universal Spectrum Identifiers (USIs)[50,52]. The spectra displayed in Figs. 3, 4, Supplementary Figs. 5, and 6 are: • Hexanoylcarnitine, C6:0: mzspec:GNPS:GNPS-LIBRARY:accession:CCMS LIB00003135669. • Hexenoylcarnitine, C6:1: mzspec:MSV000085561: 011c:scan:2864. • Benzoylcarnitine: mzspec:MSV000085561:010c:s-can:2829. • Dodecanedioylcarnitine, C12-DC: mzspec:MSV000082650: M031_48:scan:1501. • 3-Hydroxybutyrylcarnitine reference: mzs pec:GNPS:TASK-015e9e338c5649a7af6715af2be98e2f-spectra/specs_ ms.mgf:scan:1. • 3-Hydroxybutyrylcarnitine suspect: mzspec:MSV 000082049:20_51:scan:106. • 3-Hydroxyhexanoylcarnitine reference: mzspec:GNPS:TASK-015e9e338c5649a7af6715af2be98e2f-spectra/ specs_ms.mgf:scan:4. • 3-Hydroxyhexanoylcarnitine suspect: mzspec: MSV000085561:018b:scan:2609. • Apratoxin A: mzspec:GNPS:GNPS-LIBRARY:accession:CCMSLIB00000424840. • Apratoxin D: mzspec:GNPS:GNPS-LIBRARY:accession:CCMSLIB00000424841. • Apratoxin F: mzspec:GNPS:GNPS-LIBRARY:accession:CCMSLIB 00000070287. • Apratoxin C: mzspec:MSV000086109:BF9_BF9_ 02_57124.mzML:scan:722. • Apratoxin A − 30.010 Da: mzspec:MSV 000086109:BD5_dil2x_BD5_01_57213:scan:760. • Apratoxin F − 30.010 Da: mzspec:MSV000086109:BC11_dil2x_BC11_02_57176:scan: 736. • Apratoxin A − 26.015 Da: mzspec:MSV000086109:BD5_dil2x_ BD5_01_57213:scan:614. • Apratoxin A − 14.016 Da: mzspec:MSV0000 86109:BD11_BD11_02_57022:scan:591. • Azithromycin: mzspec:GNPS: GNPS-LIBRARY:accession:CCMSLIB00005434451. • 3′-O(desmethyl) azithromycin: mzspec:MSV000084132:Pos_C18_Aq7:scan:977. • Apigenin-8-C-hexosylhexoside: mzspec:GNPS:GNPS-LIBRARY:accession: CCMSLIB00004698180. • 7-O-methylapigenin-6-C-hexoside + 132.042 Da: mzspec:GNPS:TASK-38a1bd60bd094c8a97cf49d822e7f853-spec-tra/specs_ms.mgf:scan:1573560. • Apigenin-8-C-hexosylhexoside − 30.010 Da: mzspec:GNPS:TASK-38a1bd60bd094c8a97cf49d822e7f 853-spectra/specs_ms.mgf:scan:1559636. • Apigenin-8-C-hexosylhexo-side − 31.991 Da: mzspec:GNPS:TASK-38a1bd60bd094c8a97cf49d8 22e7f853-spectra/specs_ms.mgf:scan:1559563. • Apigenin-8-C-hex-osylhexoside − 46.005 Da: mzspec:GNPS:TASK-38a1bd60bd094 c8a97cf49d822e7f853-spectra/specs_ms.mgf:scan:1543689. Source data are provided with this paper.

## Code availability

Code to extract spectra from the molecular networks and compile the nearest neighbor suspect spectral library, as well as code notebooks to generate the figures and analyses presented in this manuscript are freely available on GitHub at https://github.com/bittremieux/gnps_ suspect_library under the open source BSD-3-Clause license. A permanent code archive is available on Zenodo at https://doi.org/10.5281/ zenodo.6459282. All code was implemented in Python 3.8, and uses NumPy (version 1.19.2)[53], SciPy (version 1.5.2)[54], Pandas (version 1.1.3)[55], and statsmodels (version 0.13.1)[56] for scientific data processing, Pyteomics (version 4.4.0)[57] to interface the UNIMOD repository[21], and matplotlib (version 3.5.1)[58], Seaborn (version 0.11.0)[59], spectrum_utils (version 0.3.4)[60,61], Jupyter notebooks[62], and Cytoscape (version 3.9.1.)[63] for visualization purposes.

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

## Acknowledgements

This research was supported in part by BBSRC-NSF award 2152526. This research was supported in part by National Institutes of Health awards R01 GM107550, U19 AG063744, U01AG061359, R03 CA211211, P41 GM103484, T32 HD123456. This research was supported in part by the National Institute of Aging's Accelerating Medicines Partnership for AD (AMP-AD) and was supported by NIH grants 1R01AG069901-01A1, U01AG061357, as well as by the Alzheimer Gut Microbiome Project grant 1U19AG063744. This research was supported in part by federal award DE-SC0021340 subaward 1070261-436503. This research was supported in part by the Gordon and Betty Moore Foundation through grant GBMF7622. This research was supported in part by the Intramural Research Program of National Institute of Environmental Health

Sciences of the National Institutes of Health (ZIC ES103363). WB acknowledges support by the University of Antwerp Research Fund. This research was supported in part by the National Center for Complementary and Integrative Health of the NIH under award number F32AT011475 to N.E.A. E.L.S. and T.K. acknowledge funding support from the Luxembourg National Research Fund (FNR) for project A18/BM/12341006. M.W. was partially supported by the US Department of Energy Joint Genome Institute operated under Contract No. DE-AC02-05CH11231. D.P. was supported by the Deutsche Forschungsgemeinschaft (DFG) through the CMFI Cluster of Excellence (EXC 2124). S.A.K. was supported by the Fund for Financing Science and Supporting Innovation under the Ministry of Innovative Development of the Republic of Uzbekistan. K.B.K. was supported by the National Research Foundation of Korea (NRF) grant funded by the Ministry of Science and ICT (NRF-2020R1C1C1004046). H.W.K. was supported by the National Research Foundation of Korea (NRF) grant funded by the Korean Government (MSIT) (2018R1A5A2023127). H.M.-R. acknowledges the Brazilian National Council for Scientific and Technological Development (CNPq, #142014/2018-4) and the Brazilian Fulbright Commission for the scholarships provided. L.-F.N. has been supported by the French government, through the UCA$^{J.E.D.I.}$ Investments in the Future project managed by the National Research Agency (ANR) with the reference number ANR-15-IDEX-01. J.J.J.vd.H. was supported by an ASDI eScience grant from the Netherlands eScience Center (ASDI.2017.030). C.O.D. was supported by EMBL core funds. The Alzheimer's disease metabolomics data was funded wholly or in part by the Alzheimer's Gut Microbiome Project (AGMP) NIH grant U19AG063744 awarded to R.F.K.-D. at Duke University in partnership with a large number of academic institutions. More information about the project and the institutions involved can be found at https://alzheimergut.org/meet-the-team/.

## Author contributions

P.C.D. conceptualized and supervised the work. C.O.D. and C.M.A. helped transfer and convert data from MetaboLights. W.B., M.W. and P.C.D. created the methodology to compile the nearest neighbor suspect spectral library from molecular networking data. W.B., J.M.G. and M.W. developed the software. T.H. and S.X. generated molecular formulas using BUDDY. W.B., N.E.A., S.P.T., S.A.K., A.A.A., P.W.P.G., A.M.C.R., J.M.G., A.K.J., T.K., H.M.-R., M.J.M., L.F.N., M.P., D.P., R.S., R.S., E.L.S. and J.J.J.vd.H. validated entries in the suspect spectral library and evaluated its identification performance. N.E.A., S.P.T., S.A.K., A.A.A. and P.W.P.G. provided case studies to demonstrate the utility of the tool. MW provided computational resources. C.M.A., C.O.D., M.P. and J.Z. performed data curation. A.M.C.R. acquired the acylcarnitine standards MS/MS data. W.H.G. supervised acquisition of the Moorena bouillonii MS/MS data. K.B.K., H.W.K. and H.Y. acquired the medicinal plants from the Korean Pharmacopeia MS/MS data. A.A.A. and A.M. acquired the HOMEChem MS/MS data. R.F.K.D. supervised acquisition of the ROS-MAP metabolomics data and links to ADMC and AMP-AD consortia. M.J.M., M.P., K.C.W. and J.Z. processed and prepared the ROSMAP samples and acquired the MS/MS data. W.B., N.E.A., S.P.T., A.A.A. P.W.P.G., C.M.A., M.J.M. and P.C.D. wrote the manuscript. All authors reviewed and edited the manuscript.

## Competing interests

P.C.D. consulted for DSM animal health in 2023, is an advisor and holds equity in Cybele, and is co-founder and scientific advisor and holds equity in Ometa, Arome, and Enveda, with prior approval by UC San Diego. M.W. is a co-founder of Ometa Labs LLC. A.A.A. and A.V.M. are founders of Arome Science Inc. C.M.A. is a consultant for Nuanced Health. J.J.J.vd.H. is a member of the Scientific Advisory Board of NAICONS Srl., Milano, Italy and consults for Corteva Agriscience, Indianapolis, IN, USA. R.F.K.D. is an inventor on several patents in the metabolomics field and holds founder equity in Metabolon, Chymia, and PsyProtix. The remaining authors declare no competing interests.

## Additional information

Wout Bittremieux ©[1] ✉, Nicole E. Avalon ©[2], Sydney P. Thomas[3,4], Sarvar A. Kakhkhorov ©[5,6], Alexander A. Aksenov[3,4,7,8], Paulo Wender P. Gomes ©[3,4], Christine M. Aceves[9], Andrés Mauricio Caraballo-Rodríguez[3,4], Julia M. Gauglitz[3,4], William H. Gerwick ©[2,3], Tao Huan ©[10], Alan K. Jarmusch ©[3,4,11], Rima F. Kaddurah-Daouk[12,13,14], Kyo Bin Kang ©[15], Hyun Woo Kim ©[16], Todor Kondić ©[17], Helena Mannochio-Russo ©[3,4,18], Michael J. Meehan[3,4], Alexey V. Melnik[7,8], Louis-Felix Nothias[19,20], Claire O'Donovan[21], Morgan Panitchpakdi[3,4], Daniel Petras ©[3,4,22,23], Robin Schmid ©[3,4],

Emma L. Schymanski ⑮ [17], Justin J. J. van der Hooft ⑮ [4,24], Kelly C. Weldon[3,4], Heejung Yang ⑮ [25], Shipei Xing[3,4,10], Jasmine Zemlin[3,4], Mingxun Wang[26] & Pieter C. Dorrestein ⑮ [3,4] ✉

[1]Department of Computer Science, University of Antwerp, 2020 Antwerpen, Belgium. [2]Scripps Institution of Oceanography, University of California San Diego, La Jolla, CA 92093, USA. [3]Skaggs School of Pharmacy and Pharmaceutical Sciences, University of California San Diego, La Jolla, CA 92093, USA. [4]Collaborative Mass Spectrometry Innovation Center, University of California San Diego, La Jolla, CA 92093, USA. [5]Laboratory of Physical and Chemical Methods of Research, Center for Advanced Technologies, Tashkent 100174, Uzbekistan. [6]Department of Food Science, Faculty of Science, University of Copenhagen, Rolighedsvej 26, 1958 Frederiksberg C, Denmark. [7]Department of Chemistry, University of Connecticut, Storrs, CT 06269, USA. [8]Arome Science inc., Farmington, CT 06032, USA. [9]Department of Immunology and Microbiology, The Scripps Research Institute, La Jolla, CA 92037, USA. [10]Department of Chemistry, University of British Columbia, Vancouver, BC V6T 1Z1, Canada. [11]Immunity, Inflammation, and Disease Laboratory, Division of Intramural Research, National Institute of Environmental Health Sciences, National Institutes of Health, Research Triangle Park, Durham, NC 27709, USA. [12]Department of Psychiatry and Behavioral Sciences, Duke University School of Medicine, Durham, NC 27701, USA. [13]Department of Medicine, Duke University, Durham, NC 27710, USA. [14]Duke Institute of Brain Sciences, Duke University, Durham, NC 27710, USA. [15]College of Pharmacy and Research Institute of Pharmaceutical Sciences, Sookmyung Women's University, Seoul 04310, Korea. [16]College of Pharmacy and Integrated Research Institute for Drug Development, Dongguk University, Goyang 10326, Korea. [17]Luxembourg Centre for Systems Biomedicine, University of Luxembourg, L-4367 Belvaux, Luxembourg. [18]Department of Biochemistry and Organic Chemistry, Institute of Chemistry, São Paulo State University, Araraquara 14800-901, Brazil. [19]Université Côte d'Azur, CNRS, ICN, Nice, France. [20]Interdisciplinary Institute for Artificial Intelligence (3iA) Côte d'Azur, Nice, France. [21]European Molecular Biology Laboratory, European Bioinformatics Institute (EMBL-EBI), Wellcome Genome Campus, Hinxton, Cambridge CB10 1SD, UK. [22]Interfaculty Institute of Microbiology and Infection Medicine, University of Tuebingen, 72076 Tuebingen, Germany. [23]Department of Biochemistry, University of California Riverside, Riverside, CA 92507, USA. [24]Bioinformatics Group, Wageningen University & Research, 6708 PB Wageningen, The Netherlands. [25]Laboratory of Natural Products Chemistry, College of Pharmacy, Kangwon National University, Chuncheon 24341, Korea. [26]Department of Computer Science and Engineering, University of California Riverside, Riverside, CA 92507, USA. ✉ e-mail: wout.bittremieux@uantwerpen.be; pdorrestein@health.ucsd.edu

