## [Peer Review File · Nature Communications]

Open Access Repository-Scale Propagated Nearest Neighbor Suspect Spectral Library for Untargeted MetabolomicsEditorial Note: This manuscript has been previously reviewed at another journal that is not operating a transparent peer review scheme. This document only contains reviewer comments and rebuttal letters for versions considered at *Nature Communications*.

Reviewer #2 (Remarks to the Author):

The manuscripts discussed the creation and application of a "nearest neighbor suspect spectral library" for untargeted metabolomics MS by compiling unannotated spectra from diverse public datasets. The authors propose identifying related ion species or analogous MS/MS spectra to improve annotations and using molecular networking. While the exact molecular structure of suspects is initially undetermined, they provide valuable information to interpret previously unexplored MS/MS data. The paper acknowledges that future studies may refine suspect annotations using orthogonal techniques or experimental validation.

In summary, the library addresses a crucial need in interpreting complex MS/MS spectra, and its benefits in increasing the spectrum match rate and providing structural hypotheses are significant. The authors have addressed issues raised, particularly enhancing the quality of the figures presented, both in the main text and supplementary material. Finally, it would have been great if the authors could finish the manuscript by briefly explaining their plans on how to maintain the library's accuracy and relevance over time.

Reviewer #3 (Remarks to the Author):

The authors have addressed all concerns and recommendations made by the previous reviewers.

Additionally, I recommend the following edits to the abstract to clarify the meaning of the work and resources made available to the community (I like the title and name of the library but it is important to include the word "unknown" in the abstract as it is the most common term in the field):

... derived from hundreds of millions of MS/MS spectra from published untargeted metabolomics experiments.

Entries in this library, or "suspects" were derived from unknowns or unannotated spectra that could be linked in a molecular network to an annotated spectrum.

Annotations were propagated to unknowns based on...

The nearest neighbor suspect spectral library is openly available for download as MGF or for annotation through the GNPS platform...

Please make sure all links work for publication, I could not see the library in Zenodo nor GNPS. Is it MGF or MSP format?

Minor edits:

Line 282, delete "increases".

Figure 3. Using black stars instead of yellow to indicate reference MS/MS spectra could be easier to read.

Reviewer 2

The manuscripts discussed the creation and application of a "nearest neighbor suspect spectral library" for untargeted metabolomics MS by compiling unannotated spectra from diverse public datasets. The authors propose identifying related ion species or analogous MS/MS spectra to improve annotations and using molecular networking. While the exact molecular structure of suspects is initially undetermined, they provide valuable information to interpret previously unexplored MS/MS data. The paper acknowledges that future studies may refine suspect annotations using orthogonal techniques or experimental validation.

In summary, the library addresses a crucial need in interpreting complex MS/MS spectra, and its benefits in increasing the spectrum match rate and providing structural hypotheses are significant. The authors have addressed issues raised, particularly enhancing the quality of the figures presented, both in the main text and supplementary material.

Finally, it would have been great if the authors could finish the manuscript by briefly explaining their plans on how to maintain the library's accuracy and relevance over time.

We explicitly provide the library as a community resource, available through GNPS and Zenodo for anyone to download and access. Thus, the current version is a snapshot in time that anyone can reuse. However, as public data and available reference libraries grow and new computational tools are developed, this will allow us to also update the library, as we have now mentioned in the Discussion:

As the nearest neighbor suspect spectral library was compiled from hundreds of millions of MS/MS spectra deposited to public data repositories, its creation was only possible by building on the collective efforts of the scientific community over the past decade. Consequently, besides the tangible outcome of the library itself, an important contribution of this work is demonstrating the benefit of repository-scale data analysis and integration, by unlocking patterns and insights that individual studies, in isolation, could never reveal. This represents an important phase transition that now allows us to answer questions that were previously difficult to address. Furthermore, rather than a static resource, by harnessing the continuous data reanalysis efforts of the GNPS living data system we will systematically incorporate newly deposited public data into the suspect library to continuously update its scope and applicability.

Reviewer 3

The authors have addressed all concerns and recommendations made by the previous reviewers.

Additionally, I recommend the following edits to the abstract to clarify the meaning of the work and resources made available to the community (I like the title and name of the library but it is important to include the word "unknown" in the abstract as it is the most common term in the field):

... derived from hundreds of millions of MS/MS spectra from published untargeted metabolomics experiments.

Entries in this library, or "suspects" were derived from unknowns or unannotated spectra that could be linked in a molecular network to an annotated spectrum.

Annotations were propagated to unknowns based on...

The nearest neighbor suspect spectral library is openly available for download as MGF or for annotation through the GNPS platform...

We appreciate the suggestions to better convey the approach and benefits of our work in the abstract, and have updated it accordingly. We have made one small change by omitting "as MGF" in the final sentence, as the library can be downloaded as MGF, MSP, and JSON files from the GNPS libraries portal, which would be too much technical and detailed information for the abstract.

Please make sure all links work for publication, I could not see the library in Zenodo nor GNPS. Is it MGF or MSP format?

We have carefully verified that all links in the manuscript are working. The main web pages for the suspect library are <https://gnps.ucsd.edu/ProteoSAFe/gnpslibrary.jsp?library=GNPS-SUSPECTLIST> on GNPS and <https://doi.org/10.5281/zenodo.8282733> on Zenodo. The suspect library is provided as an MGF file in the given Zenodo archive. Additionally, all spectral libraries distributed through GNPS can be downloaded as MGF, MSP, and JSON files from <https://external.gnps2.org/gnpslibrary>.

Minor edits:

Line 282, delete "increases".

We have rephrased the subsection title as follows: "Increases in MS/MS spectrum annotation provide new biomedical insights."

Figure 3. Using black stars instead of yellow to indicate reference MS/MS spectra could be easier to read.

We have changed the colors of the stars from yellow to black.